# Rolling Bearing Composite Fault Diagnosis Method Based on Enhanced Harmonic Vector Analysis

**DOI:** 10.3390/s23115115

**Published:** 2023-05-27

**Authors:** Jiantao Lu, Qitao Yin, Shunming Li

**Affiliations:** 1College of Energy and Power Engineering, Nanjing University of Aeronautics and Astronautics, Nanjing 210016, China; yin_qitao@163.com (Q.Y.); smli@nuaa.edu.cn (S.L.); 2College of Automotive Engineering, Nantong Institute of Technology, Nantong 226000, China

**Keywords:** rolling bearing, fault diagnosis, harmonic vector analysis, convolution blind source separation

## Abstract

Composite fault diagnosis of rolling bearings is very challenging work, especially when the characteristic frequency ranges of different fault types overlap. To solve this problem, an enhanced harmonic vector analysis (EHVA) method was proposed. Firstly, the wavelet threshold (WT) denoising method is used to denoise the collected vibration signals to reduce the influence of noise. Next, harmonic vector analysis (HVA) is used to remove the convolution effect of the signal transmission path, and blind separation of fault signals is carried out. The cepstrum threshold is used in HVA to enhance the harmonic structure of the signal, and a Wiener-like mask will be constructed to make the separated signals more independent in each iteration. Then, the backward projection technique is used to align the frequency scale of the separated signals, and each fault signal can be obtained from composite fault diagnosis signals. Finally, to make the fault characteristics more prominent, a kurtogram was used to find the resonant frequency band of the separated signals by calculating its spectral kurtosis. Semi-physical simulation experiments are conducted using the rolling bearing fault experiment data to verify the effectiveness of the proposed method. The results show that the proposed method, EHVA, can effectively extract the composite faults of rolling bearings. Compared to fast independent component analysis (FICA) and traditional HVA, EHVA improves separation accuracy, enhances fault characteristics, and has higher accuracy and efficiency compared to fast multichannel blind deconvolution (FMBD).

## 1. Introduction

Rolling bearings are an indispensable part of modern industry that plays an important role in wind turbine, petrochemical, metallurgical, transportation, aerospace, and other mechanical equipment [1,2]. Because rolling bearings often work in heavy loads, high temperatures, high pressures, and other harsh environments, fault damage is inevitable [3] and even causes economic losses or safety accidents. Therefore, it is of great significance to discover rolling bearing faults in time and make an accurate diagnosis.

Fault diagnosis methods for rolling bearings can be roughly divided into single fault diagnosis and composite fault diagnosis methods. Many methods have been studied for single fault diagnosis methods and could achieve good results. Commonly used signal decomposition methods include empirical mode decomposition (EMD) [4], wavelet transform (WT) [5], and variational mode decomposition (VMD) [6]. However, the above methods will have poor performance on composite fault signals with similar frequencies, and is easily interfered with by environmental noise. Compared with single faults, the frequency components of the composite fault signals of rolling bearings are more complex, and therefore, composite fault diagnosis for rolling bearings is more difficult.

Blind source separation (BSS) can estimate source signals with little prior knowledge of the signal source and transmission channel, which is widely used in signal separation and extraction [7,8,9]. According to the different mixing methods, the signal mixing model of the BSS problem can be divided into the linear instantaneous mixing model [10], the convolution mixing model [11], and the nonlinear mixing model [12].

On the BSS problem of the linear instantaneous mixed model, independent component analysis (ICA) is the most classical BSS algorithm, and fast independent component analysis (FICA) is a fast calculation method improved on ICA. As one of the most widely used algorithms, FICA is characterized by fast convergence speed and reliability. Li et al. proposed an intelligent fault diagnosis method based on FICA, which improved the problem of the insufficient fault classification and recognition ability of rolling bearings [13]. Zarzoso proposed robust independent component analysis (RICA), which takes the normalized kurtosis as the control function and optimizes the control function in each iteration of the separation filter by calculating the optimal iteration step size by an algebraic method (rather than an iterative method); by contrast, most of the other classical methods can only obtain the local mean in the search direction [14].

Although many BSS algorithms of linear instantaneous mixing models have achieved a lot of satisfactory results, their effect on mechanical signal separation is not very ideal. Under the influence of the transmission path, a convolution effect will occur between shock characteristics and modulation characteristics of composite fault signals of rolling bearings. Therefore, the convolution mixture model is more suitable for the separation of composite fault signals of rolling bearings [15]. French scholars Thomas and Deville proposed an algorithm that extends FICA to multichannel blind deconvolution, that is, fast multichannel blind deconvolution (FMBD) [16]. FMBD uses kurtosis as the control function and blind separation of signals by the successive extraction method, which is applicable to both determined and underdetermined blind source separation. Yan et al. adopted the JADE convolution BSS algorithm in the frequency domain to achieve the separation of active sonar echo signals in a marine environment full of complex noise and reverberation [17].

Independent vector analysis (IVA) is also a classical convolution BSS method, which can simultaneously estimate the mixed matrix in the frequency domain and solve the arrangement problem of separated signals to a certain extent. It is widely used in speech signals separation [18], medical imaging analysis [19], and wireless communications [20]. By considering the low-rank time-frequency structure of the source signal, IVA is further extended to an independent low-rank matrix analysis (ILRMA). ILRMA uses non-negative matrix factorization (NMF) to model the power spectrum of the source signals and achieves the best separation performance [21]. On the basis of ILRMA, Kohei Yatabe et al. proposed harmonic vector analysis (HVA) to solve the problem of BSS of convoluted mixed speech signals [22]. HVA defined time-frequency masking of speech signals with cepstral sparsity to enhance the harmonic structure and implicitly model source signals. HVA can take the spectral mode of the speech signal as ILRMA, and take each time period as IVA; therefore, HVA has the advantages of both, which makes the algorithm achieve the most advanced performance in the processing of voice and music signals.

Although HVA can achieve superior performance and efficiency for the separation of convolution mixed signals, it is still difficult to achieve ideal results when it is directly applied to the fault diagnosis of rolling bearings due to the existence of environmental noise. Wavelet threshold denoising is a good denoising method for impact signals, and it can be used to denoise the vibration signal of rolling bearings [23,24,25]. Post-processing fault signals to enhance fault characteristics is also an effective method to reduce the impact of noise, and spectral kurtosis (SK) is a good feature enhancement method. The concept of spectral kurtosis was originally proposed by Dwyer [26], and Antoni [27] systematized and normalized it, proposing a fast computation of the kurtogram based on filter banks to improve the calculation rate [28]. In the field of rolling bearing fault diagnosis, using the SK to select the demodulation frequency band with the optimal signal for filtering and then using envelope demodulation is still one of the most effective fault feature extraction methods, which can remove the influence of background noise as much as possible [29,30,31].

The purpose of this study is to introduce the HVA method from the speech signal processing field into the mechanical signal processing field, and then propose an enhanced harmonic vector analysis (EHVA) method by combining it with wavelet threshold denoising and the SK method. Compared with traditional HVA, EHVA can better deal with mechanical signals and has stronger stability in noisy environments. The above characteristics enable EHVA to achieve the ideal separation effect for composite fault signals of rolling bearings.

The organizational structure of the remaining part of this paper is as follows. Section 2 mainly introduces the theories of the proposed method, including WT denoising, determined blind source separation based on time-frequency masking, fast spectral kurtosis, and the proposed EHVA method. Section 3 verifies the effectiveness and superiority of the proposed method by the rolling bearing fault simulation experiment and comparison with the traditional BSS method. Finally, Section 4 summarizes the paper, and analyzes the advantages of the proposed method and the direction of further research.

## 2. Methodology

### 2.1. Wavelet Threshold Denoising

If function ϕ(t) satisfies condition
(1)$$C_{\phi }=\int_{-\infty }^{+\infty }\frac{{\left|\hat{\phi }\left(\omega \right)\right|}^{2}}{\left|\omega \right|}d\omega <\infty ,$$
then ϕ(t) is the generating function or basic wavelet of a wavelet, and $\hat{\phi }\left(\omega \right)$ is the Fourier transform of ϕ(t). The wavelet function is obtained by the stretching and translation transformation of the basic wavelet function, which is expressed as follows:(2)$$\phi_{a,b}\left(t\right)=\frac{1}{\sqrt{a}}\phi \left(\frac{t-b}{a}\right),$$
where a and b respectively represent the stretching and translation parameters of the wavelet transform.

The wavelet transform pair of the signal is as follows:(3)$$W_{f}\left(a,b\right)=\langle f,\phi_{a,b}\left(t\right)\rangle =\int_{-\infty }^{+\infty }f\left(t\right)\phi_{a,b}^{*}\left(t\right)dt,$$
(4)$$f\left(t\right)=\frac{1}{C_{\phi }}\int_{-\infty }^{+\infty }\frac{da}{a^{2}}\int_{-\infty }^{+\infty }W_{f}\left(a,b\right)\phi_{a,b}\left(t\right)db,$$
where * denotes conjugate. From the point of view of signal processing, wavelet threshold denoising can be regarded as a low-pass filter, but it is superior to ordinary low-pass filters because it retains the characteristics of high-frequency useful signals.

The steps of the wavelet threshold denoising method are shown in Figure 1.

### 2.2. PDS Algorithm and Time-Frequency Masking

HVA focuses on the determined BSS (DBSS) problem; that is, the number of observed signals is not less than the number of source signals. Dealing with the DBSS problem requires an assumption that the source signals are statistically independent of each other [32]. The measure of independence comes down to the following form of minimization problem:(5)$$\underset{{\left(W[f]\right)}_{f=1}^{F}}{\mathrm{Minimize}}\sum_{n=1}^{N}P_{n}(W[f]X[t,f])-\sum_{f=1}^{F}\log|\det(W[f])|$$

PDS algorithm is used to solve the minimization problem as follows:(6)$$\underset{w}{\mathrm{Minimize}}\;I(w)+J(Lw)$$
w is the quantity to be optimized, I and J are the objective functions, and L is the bounded operator. Both I and J are differentiable functions, so the optimization method based on gradient is not suitable for solving the problem. Proximity operator can be used to minimize the objective function I and J:(7)$${\mathrm{prox}}_{\mu g}[\beta ]=\underset{\delta }{\arg\min}\left[I(\delta )+\frac{1}{2\mu }\|\beta -\delta {\|}_{2}^{2}\right]$$
argmin f(x) is the variable value that minimizes f(x).

Traditional BSS methods, such as IVA, need to set an objective function. To solve this inconvenience, the time-frequency masking function was used to replace the proximity operator of the PDS-BSS algorithm [22].

Many approach operators of sparsity-induced penalty functions can be represented by threshold (or contraction) operators. The proximity operator of the 2, 1 mixture norm is represented by the group threshold operator:(8)$${\left({\mathrm{prox}}_{\lambda \|\cdot {\|}_{2,1}}[\beta ]\right)}_{n}[t,f]={\left(1-\frac{\lambda }{{\left(\sum_{f=1}^{F}{\left|\beta_{n}[t,f]\right|}^{2}\right)}^{1/2}}\right)}_{+}\beta_{n}[t,f]$$

The threshold contraction operator can be expressed as
(9)$${\left(H^{\lambda }[\beta ]\right)}_{n}[t,f]={\left(M(\beta )\right)}_{n}[t,f]\beta_{n}[t,f]$$

The group threshold operator in Equation (8) is masked with the mask as shown in Equation (10) for time-frequency masking:(10)$${\left(M_{\ell_{2,1}}^{\lambda }(\beta )\right)}_{n}[t,f]={\left(1-\frac{\lambda }{{\left(\sum_{f=1}^{F}{\left|\beta_{n}[t,f]\right|}^{2}\right)}^{1/2}}\right)}_{+}$$

According to this result, the general time-frequency masking is introduced into the PDS algorithm, and the implicit source model is defined. With this extension, any masking method can be used to estimate the separation matrix, even if the corresponding source model cannot be explicitly written as a formula [22].

### 2.3. Fast Spectral Kurtosis and Kurtogram

SK of the signal is developed from the short-time Fourier transform (STFT) of the signal, and SK is calculated by [33]:(11)$$SK=\frac{|S(t,f){|}^{4}}{|S(t,f){|}^{2^{2}}}-2,f\neq 0$$
where $\langle \left|S{\left(t,f\right)}^{n}\right|\rangle$ is the nth-order spectral moment of the signal, 〈⋅〉 is a temporal averaging operator, |⋅| is the modular operator, and S(t,f) is the complex envelope of the signal x(t) at f.

The mapping formed by SK is called a kurtogram. Antoni proposed a fast computation of the kurtogram based on 1/3 binary tree filter called fast spectral kurtosis (FK) [30]. The FK first constructs a low-pass filter h(n) with cutoff frequency $f_{c}=1/8+\delta$, where δ>0. Based on h(n), a quasi-analytic low-pass filter $h_{0}(n)$ and a quasi-analytic high-pass filter $h_{1}(n)$ are constructed, respectively [33].
(12)$$\{\begin{array}{c} h_{0}(n)=h(n)e^{j\pi n/4}\\ h_{1}(n)=h(n)e^{j3\pi n/4} \end{array}$$

Then, low-pass filter $h_{0}(n)$ and high-pass filter $h_{1}(n)$ are, respectively, used for two-fold down-sampling of filtering results layer by layer. The filter tree is shown in Figure 2.

Finally, the kurtosis of each of the above filtering results is calculated according to Equation (11). All the spectral kurtosis is combined to obtain the kurtogram quickly. More theories of the kurtogram based on 1/3 binary tree filter can be referred to in reference [28].

### 2.4. Proposed Enhanced Harmonic Vector Analysis

The process of the proposed EHVA method is as follows:

Step 1: Denoising preprocessing

The convolutional mixed observed signals with noise are denoised by the wavelet threshold denoising algorithm, and the observed signals after denoising is obtained. The process of wavelet threshold denoising is as follows: transform the original signal into the wavelet domain, and thresholding is performed in the wavelet domain to suppress small wavelet coefficients, which mainly contain random noise. The processed wavelet coefficients are used to reconstruct the signal, and the signal is obtained after suppressing the random noise.

Step 2: Obtain the fault signals by BSS

HVA is used to blind separate the observed signals after denoising. The proposed method is inspired by speech signal DBSS algorithm HVA. Speech signals are usually unsteady; that is, their characteristics change over time. Considering this characteristic, HVA converts the speech signal from the time domain to the time-frequency domain by STFT for frame processing. At the same time, the signal also has better sparsity in the time-frequency domain, which is conducive to signal separation.

Let h(t) be a time window function with center located at τ, height 1 and finite width; the STFT of the windowed signal x(t)h(t−τ) is generated as follows:(13)$${\mathrm{STFT}}_{x}(\tau ,f)=\int_{-\infty }^{+\infty }x(t)h^{*}(t-\tau )e^{-j2\pi ft}dt$$

This inner product operation maps the signal x(t) onto the time-frequency two-dimensional plane (τ,f). Here, $h^{*}(t-\tau )e^{-j2\pi ft}$ is the basis for STFT.

The time window length and its slip step length are two important parameters for STFT. Due to the inherent characteristics of human oral cavity and larynx vocalization, the best performance is achieved when the frame length is about 35 ms, and the frame overlap is about 15 ms for STFT of speech signals [34].

However, the characteristics of rolling bearing vibration signals are different from speech signals. The setting of STFT time window length and sliding step length needs to consider the fundamental frequency, sampling frequency, and signal duration. In general, when carrying out BSS based on time-frequency mask for rolling bearing signals, in order to obtain higher computational efficiency, the duration of selected observed signals is usually small. In this case, the time window length and sliding step length of STFT should also be correspondingly reduced to ensure time-frequency sparsity.

After transforming the time-domain signal into the time-frequency domain, the convolution mixing process is as follows:(14)X[t,f]=C[f]S[t,f]

The separation process is as follows:(15)W[f]X[t,f]=Y[t,f],
where $X\left(t\right)={\left[x_{1},x_{2},\cdots ,x_{M}\right]}^{T}$ is the observed signals, C is the mixed matrix of M×N, $S\left(t\right)={\left[s_{1},s_{2},\cdots ,s_{N}\right]}^{T}$ is the source signals, W is the separation matrix of N×M to be solved, and $Y\left(t\right)={\left[y_{1},y_{2},\cdots ,y_{N}\right]}^{T}$ is the separated signals. The main task of DBSS is to estimate an unmixing matrix W(f) to recover the source signals, and the core of this task is the minimization problem shown in Equation (5).

HVA first converts the log-amplitude spectrum of the observed signals into cepstrum by Fourier transform, and then carries out threshold processing on the cepstrum coefficients to enhance the dominant signal with harmonic structure and obtain the enhanced logarithmic amplitude spectrum through inverse Fourier transform. Then, the separation matrix is estimated, and blind source separation is initiated. PDS algorithm is used to minimize the objective function in BSS process, and a Wiener-like time-frequency masking function is used to replace the approach operator in the PDS algorithm to make the separation signal more independent. Finally, in order to realize the alignment of frequency scales of separated signals, a back projection method based on the principle of minimum distortion is adopted. At this point, the blind separation of observation signals is completed, and the separated signals containing fault characteristics are obtained.

Step 3: Enhance the fault characteristics

The filter structure based on 1/3 binary tree is used to filter the separated signals layer by layer, and the frequency band with the largest spectral kurtosis value of each separated signal is found, that is, the frequency band with the most concentrated fault features. Band-pass filter is used to extract the signals in the frequency band, so as to enhance the fault characteristics of separated signals.

Step 4: Fault diagnosis

The Hilbert envelope spectrum analysis is carried out for the separated signals after feature enhancement. Envelope spectrum analysis can effectively demodulate and extract the low-frequency periodic impulse signals modulated with the high-frequency natural vibration and obtain the fault characteristics which are difficult to be analyzed by spectrum analysis. According to the fault characteristics obtained, the fault diagnosis of the rolling bearing can be carried out according to its fault mechanism.

The overall flow of the proposed method is shown in Figure 3.

## 3. Experimental Procedure

To verify that this method can solve the problem of composite fault diagnosis of rolling bearings, a rolling bearing fault test platform was built. The fault experiments of different parts and degrees of rolling bearing were carried out by using the test platform, and abundant experimental data were collected to verify the proposed method.

### 3.1. Experimental Data Acquisition

As shown in Figure 4 and Figure 5, the test platform is mainly composed of a bottom plate, driving motor, bearing seat, round tube shell, shell support, and control cabinet. The rated speed of the motor is 3000 rpm.

The test platform has 3 bearing seats, with normal rolling bearings installed at both ends and test rolling bearings installed in the middle. In this experiment, one normal rolling bearing, one inner race fault rolling bearing, and one outer race fault rolling bearing were tested. The fault type was crack fault, which was made by electric discharge machining. The test rolling bearing is shown in Figure 6, and the rolling bearing parameters are shown in Table 1.

A number of sensor seats are distributed around the bearing seat on the shell. The sensor used in this test is a piezoelectric acceleration sensor, and the parameters of the sensor are shown in Table 2.

The installation position of the sensor in this experiment is shown in Figure 5 to test the axial vibration of the rotor. The experimental conditions are set as shown in Table 3.

### 3.2. Generate Observed Signals

A section of normal signal, a section of inner race fault signal, and a section of outer race fault signal were selected as the source signals, and the fault type was crack fault with a depth of 1.2 mm. The rotational speed was 1300 rpm, and the signal duration was 10 s. The timing and the speed of the signals are selected randomly. The time-domain diagram of the three source signals and their envelope spectra are shown in Figure 7.

In Figure 7, it can be seen that the inner race fault and outer race fault cause shock waveforms in the time domain. In the frequency domain of the inner race fault signal, there are obvious fault characteristics at the frequency harmonic of 116.1 Hz and its higher harmonics of 232.2 Hz, 348.3 Hz, and 464.5 Hz; their amplitudes decrease gradually. In addition, there is a fundamental frequency of 21.7 Hz on the envelope spectrum, and there is a side frequency band separated by the fundamental frequency around the fault characteristic frequency harmonic and its higher harmonics, which is a typical frequency domain characteristic of rolling bearing fault signal. The characteristic frequency harmonic of the outer race fault is 76 Hz, and its frequency domain characteristics are similar to those of the inner race fault. The difference is that the amplitudes of the characteristic frequency harmonic and its higher harmonics of the outer race fault signal are higher, while the amplitudes of the fundamental frequency and side frequency band are relatively weak. The only prominent frequency components in the normal signal envelope spectrum are the fundamental frequency and its higher harmonics, as well as some unknown high-frequency shocks.

When the source signals are collected, it already includes the influence of the transmission path and the influence of environmental noise to a certain extent. Furthermore, an FIR filter with unilateral oscillation attenuation of order R = 20 is used to convolute the source signals to simulate the process of convolution mixing of source signals by the vibration transmission path. The dimension of the filter is 3×3×20, and its time-domain diagram is shown in Figure 8. The convolution mixing process is $x(n)=\sum_{r=0}^{R-1}H(r)*s(n-r)$, where H is the causal reversible filter of order R = 20, s(n) is the source signals, and x(n) is the observed signals.

The Gaussian white noise with SNR = 0 dB is added to obtain three simulated observed signals of composite faults collected by the accelerometer, shown in Figure 9. In Figure 9, the impact of the time-domain waveforms in the observed signals is submerged, and the frequency components of the envelope spectrums are chaotic. The frequency domain of each observed signal contains different proportions of characteristic frequencies from the inner race fault signal, outer race fault signal, and normal signal, in addition to strong noise components, so it is difficult to directly diagnose all faults through observed signals.

### 3.3. Comparison of Results between Different Methods

In order to evaluate the performance of EHVA in the convolution BSS problem of rolling bearing composite fault diagnosis, this section will use the classical FICA, the advanced FMBD, the studied HVA, and the proposed EHVA, respectively, for the BSS of the observed signals generated in the previous section, and then use the envelope spectrum to compare and analyze the separation results of these three algorithms.

First, the separation effect of FICA was tested. The envelope spectrum of the separated signals is shown in Figure 10. From the envelope spectrum, it can be seen that the separated signal 1 has an inner race fault characteristic frequency harmonic of 116.2 Hz and its higher harmonics, and side frequency bands separated by a fundamental frequency, but it also has prominent unknown frequency components of 300.2 Hz and 600.3 Hz. In the envelope spectrum of separated signal 2, the amplitude of the inner race fault characteristic frequency harmonic of 116.2 Hz is lower than its side frequency. Although there are fault characteristic frequencies and their higher harmonics, the noise in the separated signal 2 is very high, and the signal-to-noise ratio is very low. In the envelope spectrum of separated signal 3, in addition to the fundamental frequency of 21.3 Hz and the inner ring fault characteristic frequency harmonic of 116.2 Hz, there is only an unknown frequency component of 228.3 Hz, and the remaining information is drowned by noise. To sum up, although the three separated signals all have inner race fault characteristics, there are many interference frequency components and high noise, so it is difficult to diagnose the fault. In addition, FICA did not isolate the outer race fault signal.

Next, FMBD is used to separate the observed signals. The envelope spectrum of the separated signals is shown in Figure 11. Due to the large storage space required for FMBD operation, only 1 s of observed signals were intercepted for the experiment. From the envelope spectrum of separated signals, it can be seen that the envelope spectrum of separated signal 1 and separated signal 2 contain the characteristic frequency harmonics of inner race faults, respectively, and the side frequency bands and higher harmonics are also visible. It can be basically determined that FMBD has separated the inner race fault signals. However, in the envelope spectrum of separated signal 1 and separated signal 2, the fault characteristic frequencies and the fundamental frequency of the separated signals are slightly different from that of the source signals (<1 Hz). This is due to the FMBD introducing a time-delay-dependent model. The amplitude of the rotor fundamental frequency of 21.7 Hz is larger than that of the fault characteristic frequency harmonic 116.4 Hz and its higher harmonics, and the amplitude of the fault characteristic frequency harmonic 116.4 Hz is smaller than that of the side frequency on the left, that is, the inner race fault signal is not separated cleanly. In the envelope spectrum of separated signal 3, only 64.8 Hz and 152.3 Hz are prominent, which is a large difference from the characteristic frequency of the outer race fault; FMBD fails to separate the outer race fault signal. On the other hand, the separation effect of FMBD also depends on whether the Gaussian characteristics of the source signals are known in advance. Therefore, although the performance of FMBD is better than FICA, there are still many shortcomings.

Then, the performance of the HVA is tested, and the envelope spectrum of the separated signals is shown in Figure 12. It can be seen that the envelope spectrum of separated signal 1 contains the inner race fault characteristic frequency harmonic of 116.2 Hz and its higher harmonics of 232.2 Hz and 348.4 Hz. However, the fundamental frequency of 21.3 Hz, side frequency of 137.5 Hz, and noise of 300 Hz have higher amplitudes, so the fault characteristics are not prominent enough. In the envelope spectrum of separated signal 2, the inner race fault characteristic frequency harmonic of 116.2 Hz and the higher harmonics of 232.2 Hz and 348.4 Hz are reflected, but the low-frequency noise is very large, which almost submerges the fault characteristics. In the envelope spectrum of separated signal 3, the outer race fault characteristic frequency harmonic of 76.1 Hz and its higher harmonics are prominent, basically meeting the requirements of fault diagnosis, but the noise is still high, and the signal-to-noise ratio is low.

HVA is a BSS algorithm with excellent performance. It was first proposed for speech signal separation. In the application of mechanical fault diagnosis based on vibration signals, the collected signals not only contain useful signals of fault information, but also contain a lot of noise. In a strong noise environment, HVA cannot be used for fault diagnosis directly. Therefore, an enhanced harmonic vector analysis algorithm is proposed by combining the wavelet threshold denoising algorithm and the fast spectral kurtosis algorithm. The method proposed in this paper has good performance in rolling bearing composite fault diagnosis.

Finally, EHVA is used for composite fault diagnosis. After wavelet threshold denoising and HVA separation, the kurtogram of the three separated signals obtained is shown in Figure 13. It can be seen from the kurtogram that the frequency band with the highest kurtosis value of each separated signal is located. Band-pass filtering is performed on this frequency band to obtain the final enhanced separated signals. The envelope spectrum of the enhanced separated signals is shown in Figure 14. It can be seen that the bandwidth of separated signals is narrowed due to band-pass filtering. In the envelope spectrum of separated signal 1, there is the prominent inner race fault characteristic frequency harmonic of 116.1 Hz and its second harmonic of 232.2 Hz. However, what is not ideal is that the amplitude of the fundamental frequency of 21.3 Hz is higher, and its bandwidth is narrow. In the envelope spectrum of separated signal 2, the inner race fault characteristic frequency harmonic of 116.1 Hz and its higher harmonics are prominent, the side frequency band is obvious, and the signal-to-noise ratio is high, which is a very ideal inner race fault signal. In the envelope spectrum of separated signal 3, the outer race fault characteristic frequency harmonic of 76.1 Hz and its higher harmonics are very prominent, and the signal-to-noise ratio is high, which is a very ideal outer race fault signal.

In this section, the blind separation performance of FICA, FMBD, HVA, and EHVA is tested by using convolutional mixed rolling bearing composite fault signals, and the separation results of the three algorithms are analyzed by using an envelope spectrum. The results show that FICA cannot achieve complete separation of fault signals. FMBD improves the signal-to-noise ratio of separated signals, and separates the inner race fault signal. However, the memory space of FMBD operation is huge, and the fault characteristic frequency of the separated signals is also prone to deviation. The separation result depends on whether the prior Gaussian characteristics of the fault signal are known. HVA can separate inner race and outer race fault signals, but the fault characteristics are not prominent enough, the noise (especially the low-frequency noise) is still large, and the signal-to-noise ratio is low. Compared with the separation result of HVA, EHVA can suppress noise and other frequency components under the condition of strong noise, retain and highlight the fault characteristics, improve the signal-to-noise ratio, and purify the fault signals.

## 4. Conclusions

In order to solve the problem of composite fault diagnosis of rolling bearings, the speech signal processing method HVA is studied.

Combining HVA with wavelet threshold denoising and fast spectral kurtosis, a rolling bearing composite fault diagnosis method, EHVA, was proposed. In the experiment part, the rolling bearing fault experiment is carried out through the rolling bearing test platform; the fault source signal is collected, the fault source signals are convolution mixed, and the Gaussian white noise is added to generate the observed signals. Then, FICA, FMBD, HVA, and EHVA are, respectively, used for blind separation of observed signals. The envelope spectrum is used to analyze and compare the performance of these three algorithms on blind separation of rolling bearing composite fault signals. The conclusions are as follows:

(1) The proposed EHVA can successfully separate all fault signals from convolutional mixed rolling bearing composite fault signals under strong noise conditions, improving the signal-to-noise ratio of separated signals;

(2) The performance of EHVA outperforms those of FICA, FMBD, and traditional HVA in the blind separation of rolling bearing composite fault signals under convolution mixed conditions. Compared with FICA and traditional HVA, EHVA could obtain a higher separation accuracy. Compared with FMBD, EHVA could have both higher separation accuracy and separation efficiency.

The proposed method still has some limitations that need to be studied in the future. First, the proposed method is mainly designed for determined BSS; that is, the number of source signals is equal to that of the sensors. Therefore, it may not perform well in underdetermined cases. Second, the proposed method can realize the separation of convolutional mixed signals, but may not effectively separate strong nonlinear mixed signals.

## Figures and Tables

**Figure 1 sensors-23-05115-f001:**
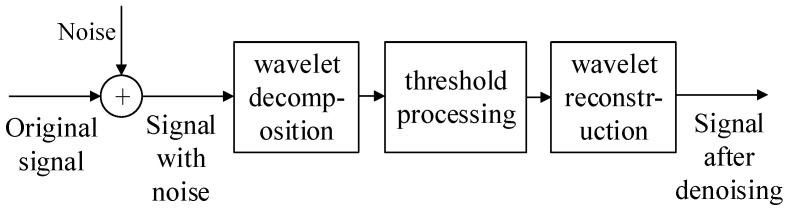
Wavelet threshold denoising process.

**Figure 2 sensors-23-05115-f002:**
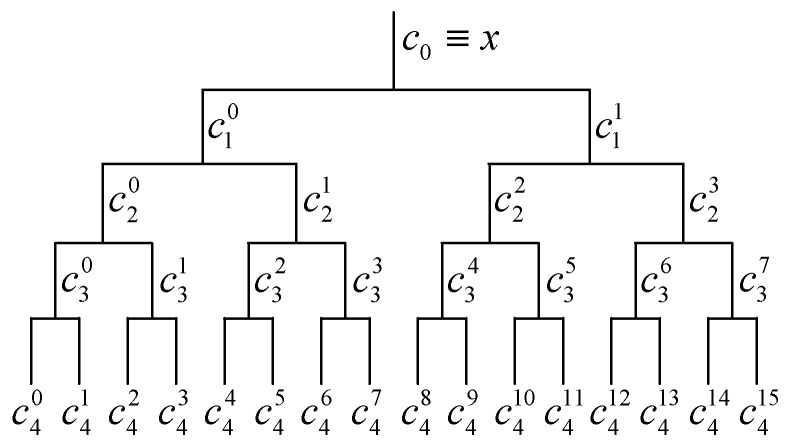
Filtering tree based on binary tree and its filtering results (after 4 iterations).

**Figure 3 sensors-23-05115-f003:**
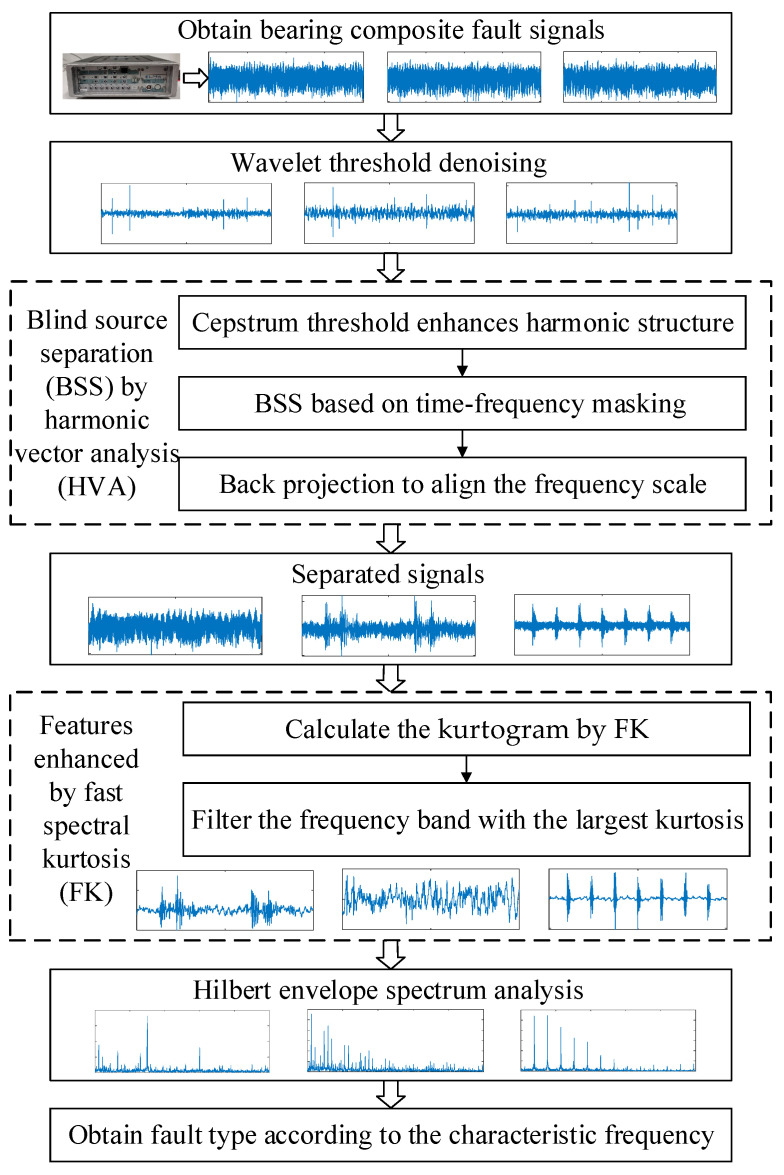
Flowchart of rolling bearing composite fault diagnosis method based on EHVA.

**Figure 4 sensors-23-05115-f004:**
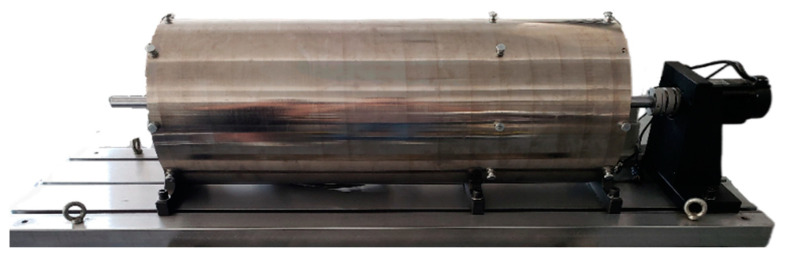
The rolling bearing test platform.

**Figure 5 sensors-23-05115-f005:**
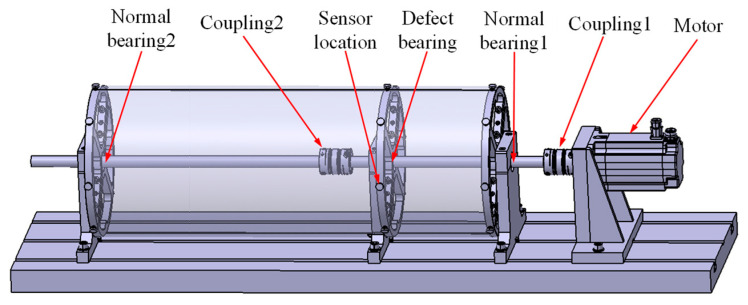
Schematic diagram of rolling bearing experimental platform.

**Figure 6 sensors-23-05115-f006:**
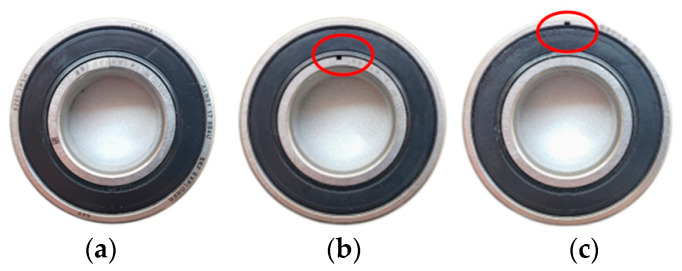
Pictures of test rolling bearing. (**a**) Normal rolling bearing. (**b**) Inner race fault rolling bearing. (**c**) Outer race fault rolling bearing. In the red circle, they are the location of fault of rolling bearing.

**Figure 7 sensors-23-05115-f007:**
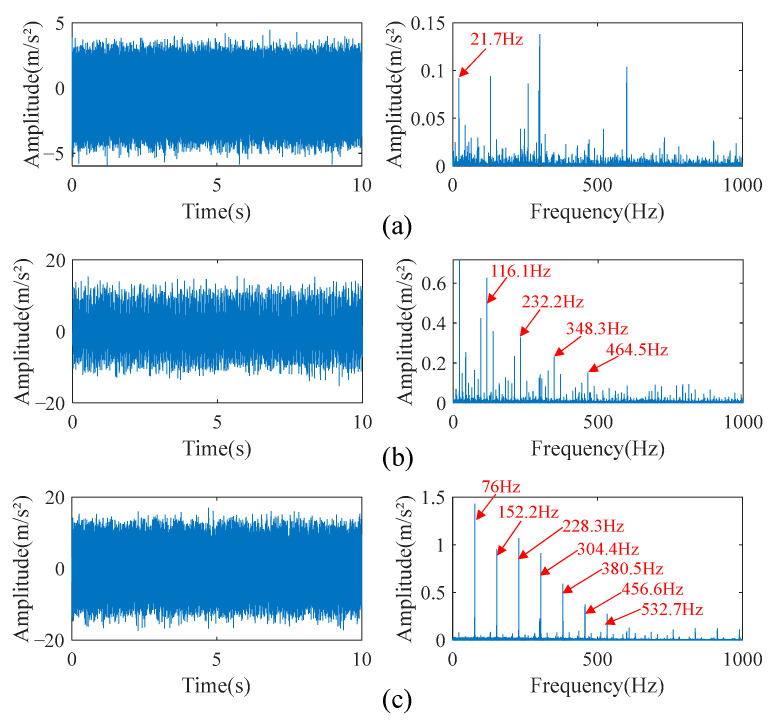
Time-domain diagram and envelope spectrum of source signals. (**a**) Normal signal. (**b**) Inner race fault signal. (**c**) Outer race fault signal.

**Figure 8 sensors-23-05115-f008:**
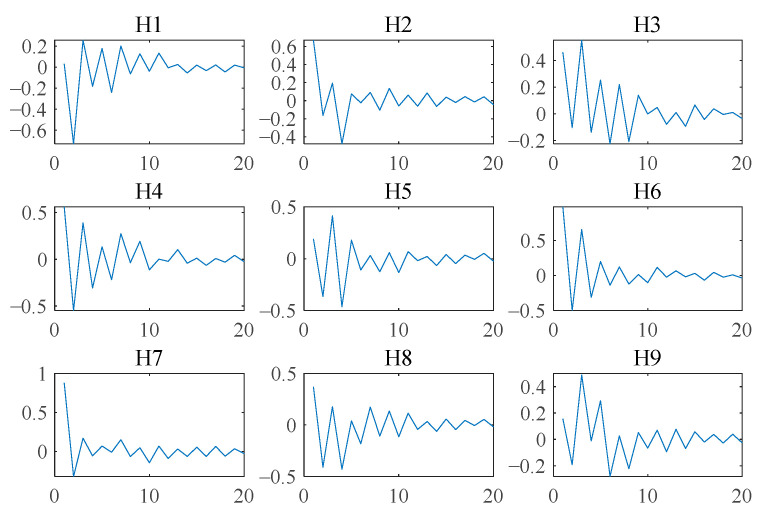
Time-domain diagram of FIR filter with unilateral oscillation attenuation of order R = 20.

**Figure 9 sensors-23-05115-f009:**
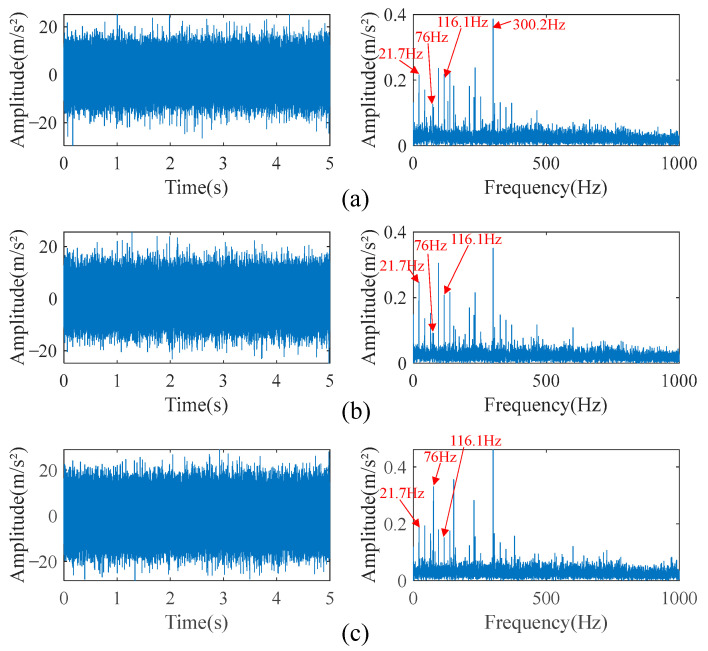
Time-domain diagram and envelope spectrum of observed signals. (**a**) Channel 1. (**b**) Channel 2. (**c**) Channel 3.

**Figure 10 sensors-23-05115-f010:**
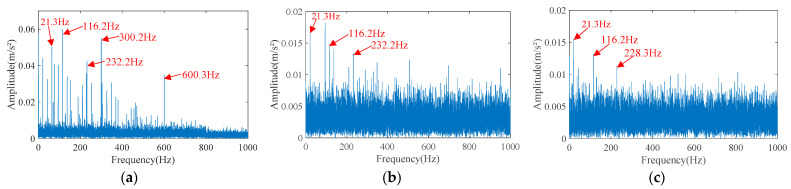
The envelope spectrum of separated signals separated by FICA. (**a**) Separated signal 1. (**b**) Separated signal 2. (**c**) Separated signal 3.

**Figure 11 sensors-23-05115-f011:**
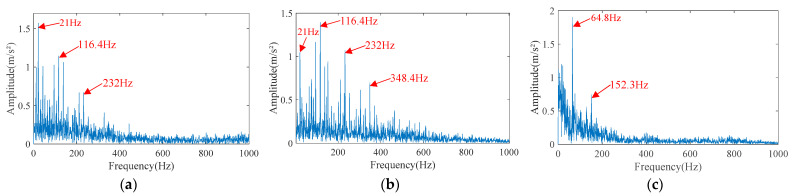
The envelope spectrum of separated signals separated by FMBD. (**a**) Separated signal 1. (**b**) Separated signal 2. (**c**) Separated signal 3.

**Figure 12 sensors-23-05115-f012:**
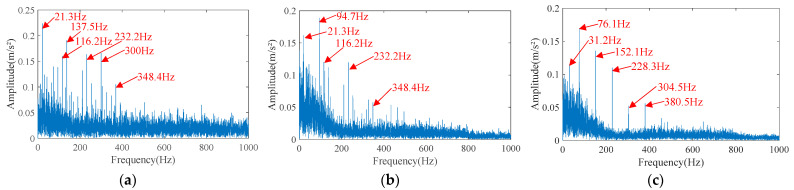
The envelope spectrum of separated signals separated by HVA. (**a**) Separated signal 1. (**b**) Separated signal 2. (**c**) Separated signal 3.

**Figure 13 sensors-23-05115-f013:**
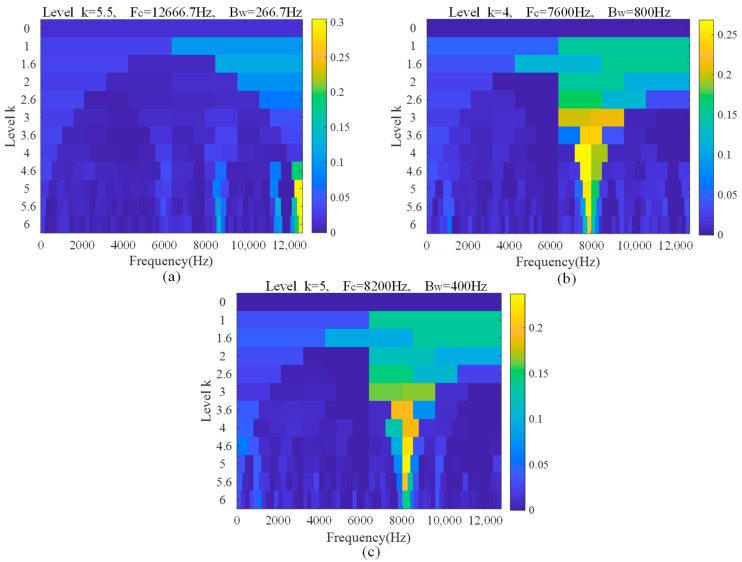
Kurtogram of three separated signals. (**a**) Separated signal 1. (**b**) Separated signal 2. (**c**) Separated signal 3.

**Figure 14 sensors-23-05115-f014:**
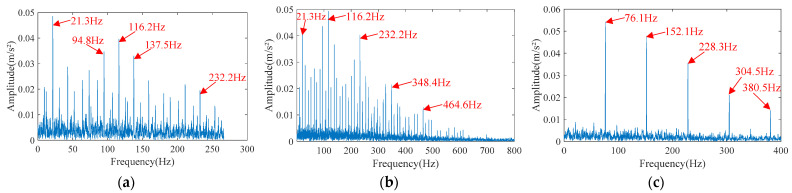
The envelope spectrum of the enhanced separated signals separated by EHVA. (**a**) Separated signal 1. (**b**) Separated signal 2. (**c**) Separated signal 3.

**Table 1 sensors-23-05115-t001:** Bearing parameters.

Features	Values
Bearing type	Deep groove ball bearing
Specification	6205-2Z
Pitch diameter (mm)	38.9
Contact angle (°)	15
Number of rollers	9
Roller diameter (mm)	7.9

**Table 2 sensors-23-05115-t002:** Sensor parameters.

Features	Values
Sensor type	YD-1182
Sensitivity (mV/ms^−2^)	10.149
Measuring range (ms^−2^)	0~500
Measuring bandwidth (kHz)	0.5~10
Output mode	Top L5

**Table 3 sensors-23-05115-t003:** Experimental conditions.

Features	Values
Sampling frequency (kHz)	25.6
Sampling time (s)	20
Accelerometer	Piezoelectric accelerometer YD-1182
Fault crack depth (mm)	0.2, 0.6, 1.2
Rotor speed (rpm)	300, 500, 796, 1018, 1300, 1600, 1837, 2012, 2300, 2988

## Data Availability

The data presented in this study are available on request from the corresponding author.

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
