# Peer review of "Rolling Bearing Composite Fault Diagnosis Method Based on Enhanced Harmonic Vector Analysis"

_sensors, 2023, doi:10.3390/s23115115_

Round 1

Reviewer 1 Report

1.      Acronyms should be defined the first time they appear in each of three sections: the abstract; the main text; the first figure or table. FICA and FMBD are not defined in the abstract and Figure 3 also has many undefined acronyms, please correct them and heck in the full text.

2.      The basic theory of non-original content of fast spectral kurtosis can be more condensed. Readers may be referred to relevant references if necessary.

3.      There are some extra blank lines in Section 2.4. Please check and modify it.

4.      The ordinals in Figure 6 are not aligned with the corresponding subgraphs. Please check and adjust them.

5.      In 3.1 Experimental data acquisition, add the description of sensor types and parameters.

No

Reviewer 2 Report

The article proposes a composite fault diagnosis method for rolling bearings, which has certain reference significance for bearing fault diagnosis. However, there are still several issues that need further explanation or modification:

1.     In the lines 142-145, there is incomplete display of formulas in the PDF version. Please pay attention to completeness during the format conversion process.

2.     In the lines 211 to 213, please note consistency in formula form.

3.     In the line 224, no period at the end of the sentence.

4.     In section 3.2, there are some unclear explanations on how composite faults are constructed in generating observed signals, such as how many composite faults are formed, whether through experiments or signal superposition methods, and the basis for construction.

5.     In the line317-320,Do the three simulated observed composite fault signals all contain the normal segment signal, the outer ring fault signal, and the inner ring fault signal? Then what is the difference between the three simulated observed composite fault signals?

6.     In the line 335-338,In Figure 9 (a), the inner circle fault characteristic frequency harmonic is 116.2Hz, but here it is written as 116.1Hz, which is not consistent with the data labeled in the figure.

7.     In the line 350-351,why choose to separate out three separated signals, rather than two or four, the three separated signals are intended to represent the normal signal, inner race fault signal, outer race fault signal three cases?

8.     In the line 417-418, from all the above methods combined, the three signals separated, Figure (a) and Figure (b) represents the separated inner race fault signal, Figure (c) represents the separated outer race fault signal, with Figure 9 annotations, why choose to separate the three signals?

9.     In the section 3.3comparison of results between different methods,do the three separated signals correspond to the three channel signals in the simulated observed composite fault signals, or is one simulated observed composite fault signal separated from three separated signals?
